# The impact of cyberbullying on mental health outcomes amongst university students: A systematic review

**Aahan Arif** [1,2], **Muskaan Abdul Qadir**[1,2], **Russell Seth Martins**[3], **Hussain Maqbool Ahmed Khuwaja** [4]*

**1** Medical College, Aga Khan University Hospital, Karachi, Pakistan, **2** Research & Development Wing, Society for Promoting Innovation in Education, Aga Khan University Hospital, Karachi, Pakistan, **3** Department of Surgery, Division of Thoracic Surgery, JFK University Medical Center, Hackensack Meridian School of Medicine, Edison, New Jersey, United States of America, **4** School of Nursing and Midwifery, Aga Khan University Hospital, Karachi, Pakistan

☯ These authors contributed equally to this work.
* hussainmaqbool.khuwaja@gmail.com

**Data Availability Statement:** All data generated and analyzed during this study are included in this published article.

## Abstract

Cyberbullying is increasingly prevalent globally, particularly among young individuals. Cybervictims may be at an increased risk of adverse psychological outcomes. This systematic review aims to summarize the mental health effects of cyberbullying among college and university students. A systematic search of PubMed, Cochrane, and Embase databases was performed to identify studies reporting mental health effects of cybervictimization among college/university students until April 15, 2023. Risk of bias assessment was conducted using the National Institute of Health (NIH) tool. The review is registered on PROS-PERO (CRD42023429187). Thirty-two studies involving 29,593 students were included. Depression showed a significant association with cyber-victimization in 16/20 studies (prevalence: 15–73%). Anxiety was significant in 12/15 studies (27–84.1%), stress in 3/3 studies (32–75.2%), and suicidal behavior in 4/9 studies (2–29.9%). Cybervictimization weakly but significantly correlated with lower self-esteem in 4 out of 6 studies (r = -0.152 to -0.399). Fear of perpetrators was reported in 2 out of 2 studies (12.8–16%), while decreased academic concentration/productivity was found in two studies (9–18%). Cybervictims were more likely to engage in substance abuse (adjusted odds ratio: 2.37 [95% confidence interval: 1.02–5.49]; p = 0.044). The majority of articles were of good quality (22/32). This review demonstrates a high prevalence of adverse mental health outcomes among cybervictims, including depression, anxiety, stress, and suicidal behavior. Based on these findings, we recommend that institutions of higher education worldwide introduce zero tolerance policies against cyberbullying, implement screening processes to identify affected students, and provide psychological therapy within their institutions.

**Funding:** The authors declare that they have no source of funding.

**Competing interests:** The authors declare that they have no competing interests.

## Introduction

With the increasing popularity of social media in the past decade, bullying has found its way into the digital sphere. Cyberbullying, defined as bullying via electronic means [1], has become increasingly prevalent across the globe, with more than half of adults in the United States (US) with access to the internet having had a cyberbullying experience [2].

Bullying is recognized as a global public health issue, since individuals exposed to bullying are more likely to develop mental health problems [3]. While cyberbullying may be seen as an extension of traditional bullying, its impact on mental health has the potential to be far more devastating given the anonymity and lack of supervision in the cyberspace [4]. This anonymity enables bullies to hide behind online aliases and continue to inflict psychological distress on their victims. In addition, an online platform magnifies the reach of a cyberbully, with subsequent mental health repercussions being potentially more far reaching than those observed in traditional bullying [5]. The issue of cyberbullying is of greater concern among university students as they spend considerable time on the internet and social media services and are thus at higher risk of cybervictimization [6]. Youth is a pivotal time in one's development, as physical, emotional, and social changes during this period of transition can predispose individuals to developing mental health problems [7].

While cyberbullying has been explored amongst adults in general, the burden of this phenomena and consequent psychological distress remains to be comprehensively investigated amongst college and university students who represent a high-risk population [8]. Thus, in this systematic review, we aim to summarize the mental health outcomes associated with cybervictimization amongst university students.

## Methods

The review was conducted in accordance with the PRISMA guidelines [9]. This systematic review is registered on PROSPERO (CRD42023429187).

### Search strategy

We conducted a systematic search of MEDLINE, Cochrane, Embase, and Google Scholar on 15/04/2023 to identify articles discussing cybervictimization and mental health outcomes at the undergraduate or postgraduate university level (**Fig 1**). The search string was divided into three components as follows: cyberbullying and associated synonyms, mental health and associated synonyms, and university and associated synonyms (**S1 Search strings**).

### Selection criteria

The following were the inclusion criteria required for articles to be shortlisted:

- Population: Students currently enrolled in undergraduate or postgraduate educational institutions.

- Intervention: Cyberbullying during undergraduate or postgraduate education.

- Control: No experience of cyberbullying.

- Outcome: Any adverse mental health outcomes.

### Study selection

Articles identified using the search string were imported into Google Sheets, and duplicates were removed. The titles and abstracts of all the articles were screened as per the eligibility

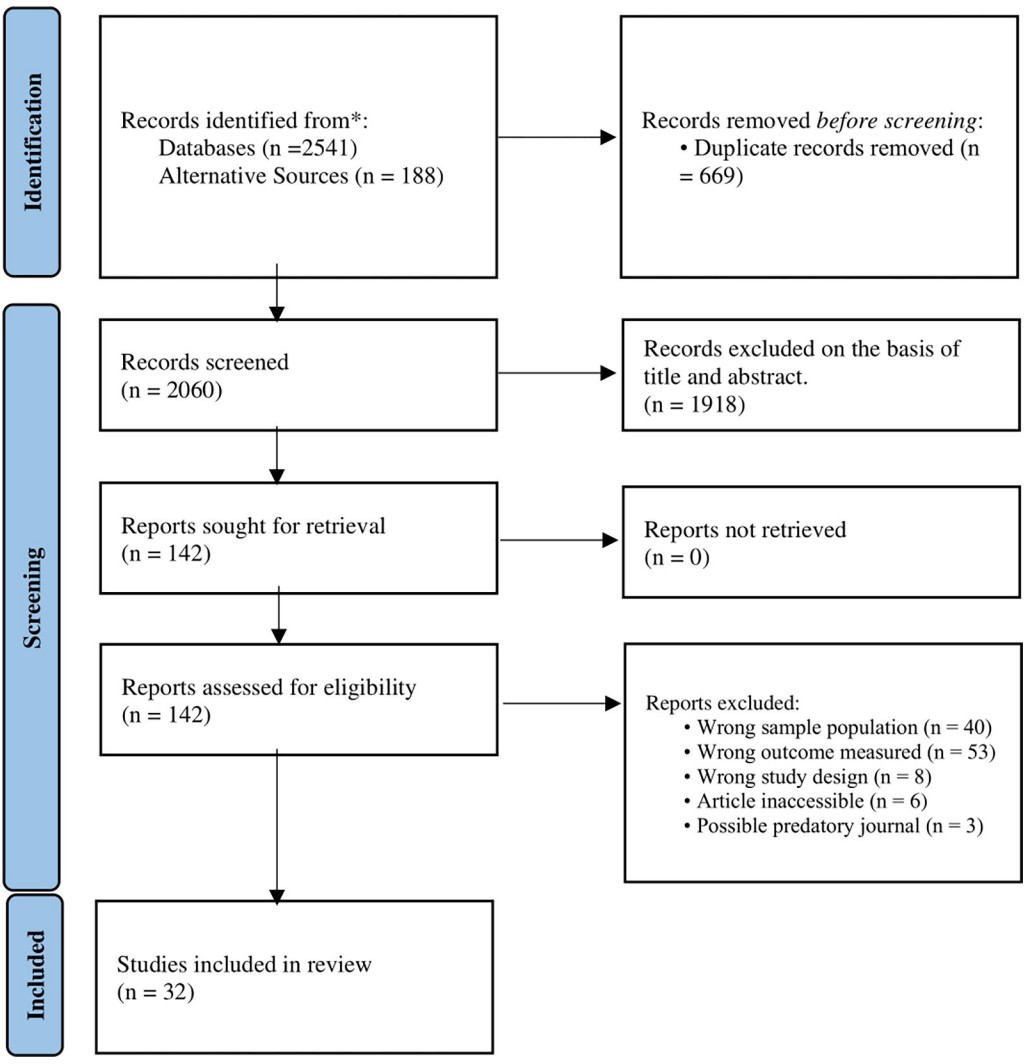

**Fig 1. This figure is the PRISMA used in synthesis of the systematic review.**

criteria by two members (AA and MAQ) of the research team. In cases of ambiguity, consensus was reached by a third member (RSM) of the team and the senior author (HMAK). Following this, a final list of articles meeting the selection criteria was created.

## Quality assessment

Quality assessment was performed using the National Institutes of Health (NIH) tool for Quality Assessment of Observational Cohort and Cross-Sectional Studies [10]. Two authors (AA and MAQ) independently assessed the quality of each individual article. Criteria 9, 11, and 14 of the NIH tool [10] were deemed "major" criteria, while criteria 10, 12, and 13 were considered not applicable as they were for cohort studies. The remaining criteria were deemed "minor" criteria. Articles where all three major criteria were mentioned were deemed to be of good quality. Articles with two major criteria mentioned were deemed of fair or good quality based on minor criteria as per the authors' discretion. The remaining articles were deemed poor. Following this, any discrepancies were discussed, and consensus was resolved in consensus with a third author (RSM) and the senior author (HMAK).

### Data extraction and management

Data was extracted to Google Sheets on 10/06/2023 by two authors (AA and MAQ). Extracted parameters included author name, year of publication, country of publication, sample size, age of participants, cyberbullying measurements, and mental health outcomes measured. Moreover, statistical measures used to evaluate mental health outcomes as well as their associated results were also extracted.

### Statistical analysis

Publication bias was assessed using R. version 4.3.0. Outcomes where prevalences was available for at least three articles were included. To evaluate outcomes, the Free-man Turkey double arcsine transformation was utilized, with subsequent generation of funnel plots (**S1–S7 Figs**). Funnel plots were then tested for statistical evidence of bias using the Begg funnel plot test.

## Results

### Study characteristics

A total of 32 articles, spanning 2010–2023, met the selection criteria and were included in the systematic review (**Fig 1**). Characteristics of the included studies are shown in **Table 1**. A total of 29,593 students, with mean ages ranging from 19.55–23.79 years [11,12], were included across the 32 articles. The commonest countries of origin of the articles were the US [11,13–24] (n = 13), Spain [25–28] (n = 4), and Canada [12,24,29] (n = 3). Sample sizes ranged from 107 to 6740 participants [22,30]. Of the 32 articles, 23 utilized pre-existing questionnaires from the literature [11,12,16–19,22–28,31–35] while the remainder used self-developed tools. The percentage of male students ranged from 19.8–68.7% [25,36].

### Prevalence of cybervictimization

The prevalence of cybervictimization ranged from 6.9% to 84.3% [12,24]. When comparing prevalence between genders, 4 articles [19,30,32,35] found cybervictimization to be significantly more common amongst female students.

### Mental health outcomes

The commonest reported mental health outcomes were depression [11,12,14–19,21,23,26,27,31–35,37–39] (n = 20 articles), anxiety [16–20,23,24,26,27,31–33,38–40] (n = 15 articles), and suicidality [16,21,27,29,34,36,37,41,42] (n = 9 articles). Among the 20 articles discussing depression, 16 articles [11,12,14,16–19,23,26,27,33–35,37–39] found that cybervictimization was significantly associated with the development of depression (**Table 2**). Cybervictimization was significantly associated with the development of anxiety in 12 articles [16–20,23,26,27,33,38–40] and suicidality in 4 [16,27,34,37] articles. Cybervictimization was found to be associated with greater psychological symptoms (e.g., distress, excessive rumination, loneliness, etc.) in 4 studies [16,26,31,36], lower self-esteem in 4 articles [12,17,28,39], higher stress in 3 studies [27,33,38], internet/social media addiction in 2 articles [39,40], and overall poorer mental well-being in 2 articles [32,33]. A comprehensive account of the statistical measures used in each individual study is shown in (**S1 Table**).

### Quality assessment

Out of the 32 articles, 22 were deemed to be of good quality [11,14–17,20,22,23,25–28,31,33–40,42], 8 of fair [12,13,18,19,24,29,30,32], and 2 of poor quality [21,41] (**S1 Table**). Criteria 1

**Table 1. Characteristics of included studies, mental health outcomes recorded, and study tools used.**

| # | Author | Year | Country | Sample size | Age (in years) * | Cyberbullying measurements | Mental health outcome measured |
|---|--------|------|---------|-------------|------------------|----------------------------|--------------------------------|
| 1 | Albikawi et al. [37] | 2023 | Saudi Arabia | N = 179 100% female | 20.80 ± 1.62 | • Self-designed questionnaire | • Self-esteem via RSES [38] • Anxiety via SAS [39] • Internet addiction via CIAS [40] • Depression via WHO-5 [41] |
| 2 | Alrajeh et al. [35] | 2021 | Qatar | N = 836 • 81.5% females • 18.5% males | 18–24 | • RCBI-II [42] | • Depression via PHQ-9 [43] |
| 3 | Anis-ul-Haque et al. [32] | 2018 | Pakistan | N = 508 • 68.5% female • 31.5% male | 18–25 | • Adapted from Rey et al. [44] | • Depression via DASS-21 [45] • Anxiety via DASS-21 [45] • Psychological distress via DASS-21 [45] • Mental wellbeing via WEMWB [46] |
| 4 | Arafa et al. [30] | 2017 | Egypt | N = 6740 • 34% male • 66% female | 18–22 | • Self-designed questionnaire | • Psychological consequences of cyberbullying via self-designed questionnaire |
| 5 | Beran et al. [24] | 2012 | Canada & US | N = 1,368 • 50.6% male • 49.4% female | 17–25 | • Adapted from the university harassment survey, Beran et al. [47] | • Emotional behavior and impact via university harassment survey, Beran et al. [47] |
| 6 | Cenat et al. [36] | 2019 | France | N = 4 626 • 19.8% male • 90.2% female | 15–23 | • Self-designed questionnaire | • Psychological distress via self-designed questionnaire • Suicidal Ideation via self-designed questionnaire • Emotional abuse from parents via self-designed questionnaire |
| 7 | Chu et al. [34] | 2022 | China | N = 1509 • 45.5% male • 54.5% female | 16–25 | • Chinese version of Revised Cyberbullying Inventory [48] | • Depression via Chinese version of CES-D [49] • Core self-evaluation via Chinese version of Core Self-Evaluation Scale [50] • Suicidal ideation via the BSI-CV [51] |
| 8 | Elipe et al. [25] | 2015' | Spain | N = 636 • 68.7% male • 31.3% female | 18–25 | • ECIPQ [52] | • Emotional impact • Perceived emotional intelligence via Trait Meta-Mood Scale-24 [53] |
| 9 | Faucher et al. [29] | 2014 | Canada | N = 1925 • 26% male • 74% female | Not mentioned | • Self-designed questionnaire | • Psychological distress via self-designed questionnaire • Anxiety via self-designed questionnaire • Depression via self-designed questionnaire • Suicidal Ideation via self-designed questionnaire |
| 10 | Feinstein et al. [11] | 2014 | US | N = 565 • 35.3% male • 64.7% female | 17–22 | • Adapted from Internet Harassment Experiences questionnaire [54] | • Depression via the depression subscale of the short version of the DASS-21 [45] • Rumination via the 22 item Ruminative Responses Scale (RRS) [55] |
| 11 | Felipe-Castaño et al. [26] | 2019 | Spain | N = 1108 • 41% male • 59% female | 17–24 | • Scale of Victimization Through Internet via Buelga et al. [56] • Scale of Aggression Through Internet via Buelga et al. [57] | • Depression via Sandin et al. [58] • Hostility via Sandin et al. [58] • Interpersonal Sensitivity via Sandin et al. [58] • Anxiety via Sandin et al. [58] • Psychoticism via Sandin et al. [58] • Obsession-Compulsion via Sandin et al. [58] • Phobic Anxiety via Sandin et al. [58] • Paranoid Ideation via Sandin et al. [58] |
| 12 | Giumetti et al. [23] | 2022 | US | N = 317 • 19.2% males • 80.1% females • 0.6% preferred not to answer | 19–31 | • WCM [59] | • Depression via DASS [45] • Anxiety via DASS [45] • Alcohol use via single question • Helping Behavior adapted from Lee et al. |
| 13 | Huang et al. [60] | 2021 | China | N = 897 • 43.8% male • 56.2% female | 15–25 | • Self-designed questionnaire | • Self-esteem via RSES [38] • Anxiety via SAS [39] • Internet/social media addiction via CIAS-R [61] |

*(Continued)*

**Table 1.** (Continued)

| # | Author | Year | Country | Sample size | Age (in years) * | Cyberbullying measurements | Mental health outcome measured |
|---|---|---|---|---|---|---|---|
| 14 | Kaur et al. [62] | 2020 | Malaysia | N = 270<br>•24.4% male<br>•75.6% female | 21.98 ± 1.54 | • Self-designed questionnaire | • Depression via DASS-21 [45]<br>• Anxiety via DASS-21 [45]<br>• Stress via DASS-21 [45] |
| 15 | Khine et al. [63] | 2020 | Myanmar | N = 412<br>• 67.2% male<br>• 32.8% female | 18–22 | • Self-designed questionnaire | • Substance abuse via self-designed questionnaire<br>• Concentration and cognitive difficulties via self-designed questionnaire<br>• Suicidal Ideation via self-designed6questionnaire |
| 16 | Kowalski et al. [22] (Study 2) | 2012 | US | N = 107<br>• 37.4% male<br>• 62.6% female | 18–33 | • Adapted from Cortina et al. [64]<br>• Additional five questions that asked about further details of their cyberbullying experiences at work | • Psychological burnout via the Shirom et al. [65] |
| 17 | Kraft et al. [21] | 2010 | US | N = 471<br>• 24% male<br>• 76% female | 80% participants under 25 | • Self-designed questionnaire | • Psychological distress via self-designed questionnaire<br>• Depression via a self-designed questionnaire<br>• Suicidal Ideation via a self-designed questionnaire |
| 18 | Lai et al. [66] | 2017 | Malaysia | N = 712<br>• 43.5% male<br>• 56.5% female | 19–24 | • Self-designed questionnaire | Impact of cyberbullying via self-designed questionnaire |
| 19 | Lam et al. [20] | 2022 | US | N = 486<br>• 32% male<br>• 67% male | 17–47 | • Self-designed questionnaire | • Anxiety via SIAS [67]<br>• Social comparison via INCOM [68] |
| 20 | Lee et al. [19] | 2020 | US | N = 356<br>• 69.7% female<br>• 30.3% male | 19–25 | • The Cyberbullying Victimization Scale [69]<br>• The Cyberbullying Perpetration Scale [69] | • Depression via 14-item CES-D [70]<br>• Anxiety via the 14-item DASS [45] |
| 21 | Lindsay et al [18] | 2015 | US | N = 342<br>• 24.6% male<br>• 75.4% female | 16–28 | • Adapted from Finn et al. [71] | • Depression via a single yes/no question<br>• Anxiety via a single yes/no question |
| 22 | Martínez-Monteagudo et al. [27] | 2020 | Spain | N = 1282<br>• 53.7% female<br>•46.3% male | 18–46 | • ECIPQ [44] | • Depression via DASS-21 [72]<br>• Anxiety via DASS-21 [72]<br>• Psychological distress via DASS-21 [72] |
| 23 | Medrano et al. [73] | 2017 | Mexico | N = 303<br>• 59.1% female<br>• 40.9% male | 18–24 | • Cyberbullying Victimization Questionnaire (CBQ-V) [74] | • Depression via CES-D Short Version [75]<br>• Suicidal Ideation via The negative suicidal ideation scale of the Inventory of Positive and Negative Suicide Ideation [76] |
| 24 | Musharraf et al. [33] | 2018 | Pakistan | N = 508<br>• 68.5% female<br>• 31.4% male | 18–25 | • Adapted from Del Rey et al. [44] | • Depression via DASS-21 [45]<br>• Anxiety via DASS-21 [45]<br>• Psychological distress via DASS-21 [45]<br>• Mental well-being via WEMWB [46] |
| 25 | Na et al. [17] | 2015 | US | N = 121<br>• 38.6% female<br>• 61.4% male | 18–21 | • Adapted from Patchin et al. [77] | • Depression via DASS-21 [78]<br>• Anxiety via DASS-21 [78]<br>• Self-esteem via RSES [38] |
| 26 | Sam et al. [31] | 2018 | Ghana | N = 844<br>• 57.1% male<br>• 42.9% female | 14–23 | • Adapted from Menesini et al. [79] | • Personality traits of neuroticism via NEO-FFI [80]<br>• Self-esteem via RSES [38]<br>• Anxiety via scale taken from Berry et al [81].<br>• Depression via Berry et al [81].<br>• Psychosomatic symptoms via Berry et al [81]. |
| 27 | Schenk et al. [16] | 2012 | US | N = 799<br>• 71.6% female<br>• 28.4% male | 18–22 | • Internet Experiences Questionnaire [16] | • Depression via SCL-90-R [82]<br>•Anxiety via SCL-90-R [82]<br>• Hostility via SCL-90-R [82]<br>• Paranoid ideation via SCL-90-R [82]<br>• Psychoticism via SCL-90-R [82]<br>n via SBQ-R [83] |

*(Continued)*

**Table 1.** (Continued)

| # | Author | Year | Country | Sample size | Age (in years) * | Cyberbullying measurements | Mental health outcome measured |
|---|--------|------|---------|-------------|------------------|----------------------------|--------------------------------|
| 28 | Selkie et al. [15] | 2015 | US | N = 265<br>• 100% female | 18–25 | • A single yes/no MCQ asking about participating in or experiencing cyberbullying | • Depression via PHQ-9 [43]<br>• Alcohol use via The Alcohol Use Disorder Identification Test [84] |
| 29 | Snaychuk et al. [12] | 2020 | Canada | N = 127<br>• 28% male<br>• 72% female | 16–31 | • Adapted from Powell et al. [85] | • Self-Esteem via RSES [38]<br>• Social Support via Zimet et al. [86]<br>• Perceived Control via e Schooler et al. [87]<br>• Depression via Beck et al. [88] |
| 30 | Wright et al. [14] | 2023 | US | N = 127<br>• 40% male<br>• 60% female | 19.98 ± 0.89 | • Self-designed questionnaire | • Depression via CES-D [70] |
| 31 | Yubero et al. [28] | 2017 | Spain | N = 243<br>• 32% male<br>• 68% female | 19–40 | • Escalas de victimización a traves de Internet—Internet Victimization Scales [89] | • Self-esteem via RSES [38]<br>• Loneliness via UCLA Loneliness Scale [90]<br>• Peer acceptance via Perceived Acceptance Scale [91] |
| 32 | Zalaquett et al. [13] | 2014 | US | N = 604<br>• 75.9% females<br>• 24.6% males<br>• 0.5% did not report their gender | 21–59 | • Self-designed questionnaire | • Psychological distress via self-designed questionnaire. |

Key

*: Age was reported as ranges where possible. If not possible means and standard deviations or given information was reported.

*ECQIP*: European Cyberbullying Intervention Project Questionnaire.

*DASS*: Depression, Anxiety, and Stress Scale.

*RRS*: Ruminative Responses Scale.

*CES-D*: Center for Epidemiologic Studies Depression Scale.

*SCL-90-R*: Symptom Checklist-90-Revised.

*RSES*: Rosenberg Self-Esteem Scale.

*PHQ-9*: Patient Health Questionnaire-9.

*SBQ-R*: Suicidal Behaviors Questionnaire-Revised.

*WEMWB*: Warwick–Edinburgh Mental Well-being Scale.

*SAS*: Self-anxiety Scale.

*CIAS*: Chen Internet Addiction Scale.

*WHO-5*: WHO-Five Well-Being Index.

*RCBI-II*: Revised Cyberbullying Inventory Scale.

*BSI-CV*: Beck Scale for Suicidal Ideation- Chinese Version.

*WCM*: Workplace Cyberbullying Measure.

*SIAS*: Social Interaction Anxiety Scale *INCOM*: Iowa-Netherlands Comparison Orientation Measure.

11–42, 4, 11–19, 21–31, 33, 34, 36–42, 9 [11–20,23–39,42], 11 [11,12,14–20,22,23,25–28,30–40,42], and 14 [11,14–17,20,23–28,31,33–40,42] of the NIH tool [10] were met by more than 50% of articles. Criteria 9 referred to the validation of the study exposure questionnaire [10] and was met by 28 articles [11–20,23–39,42]. Criteria 11 referred to the validation of the study outcome questionnaire [10] and was met by 27 articles [11,12,14–20,22,23,25–28,30–40,42].

## Publication bias

A total of seven outcomes (anger, anxiety, depression, lack of concentration/loss of productivity, sadness, stress, and suicidality) across eleven articles were analyzed for potential publication bias [13,16,18,21,24,27,29,30,33,41,42]. Strong evidence of publication bias was found

**Table 2. The association of cybervictimization with mental health outcomes.**

| Key Factors | Reference | Results |
|---|---|---|
| **Depression** | **Albikawi et al. [37]** | Cyberbully victims were more likely to have experienced depression. * |
| | **Alrajeh et al. [35]** | Cyberbully victims were more likely to experience depressive symptoms. * |
| | **Chu et al. [34]** | Weakly significant positive correlation between cybervictimization and depression. * |
| | **Feinstein et al. [11]** | Cyberbully victims experienced significantly greater depression. * |
| | **Felipe-Castaño, E. et al. [26]** | Weakly significant positive correlation between cybervictimization and depression. * |
| | **Giumetti et al. [23]** | Weakly significant positive correlation between cybervictimization and depression. * |
| | **Kaur et al. [62]** | Cyberbully victims were more likely to have experienced depression. * |
| | **Lee J [19]** | Weakly significant positive correlation between cybervictimization and depression. * |
| | **Lindsay et al. [18]** | Cyberbully victims experienced significantly greater depression. * |
| | **Martinez-Monteguado et al. [27]** | Cyberbully victims were more likely to have experienced depression. * |
| | **Medrano et al. [73]** | Weakly significant positive correlation between cybervictimization and depression. * |
| | **Musharraf et al. [33]** | Weakly significant positive correlation between cybervictimization and depression. * |
| | **Na et al. [17]** | Weakly significant positive correlation between cybervictimization and depression. * |
| | **Sam et al. [31]** | Insignificant difference between cyberbullying victims and non-victims. |
| | **Schenk et al. [16]** | Weakly significant positive correlation between cybervictimization and depression. * |
| | **Selkie et al. [15]** | Cyberbully victims were not more likely to have experienced depression. |
| | **Snaychuk, L.A. et al. [12]** | Cyberbully victims experienced significantly greater depression. * |
| | **Wright et al. [14]** | Weakly significant positive correlation between cybervictimization and depression. * |
| **Anxiety** | **Albikawi et al. [37]** | Cyberbully victims were more likely to have experienced anxiety. * |
| | **Felipe-Castaño, E. et al. [26]** | Weakly significant positive correlation between cybervictimization and anxiety. * |
| | **Giumetti et al. [23]** | Weakly significant positive correlation between cybervictimization and anxiety. * |
| | **Huang et al. [60]** | Cyberbully victims in social media were more likely to have experienced anxiety. * |
| | **Kaur et al. [62]** | Cyberbully victims were more likely to have experienced anxiety. * |
| | **Lam et al. [20]** | Weakly significant positive correlation between cybervictimization and anxiety. * |
| | **Lee J [19]** | Weakly significant positive correlation between cybervictimization and anxiety. * |
| | **Lindsay et al. [18]** | Cyberbully victims experienced significantly greater anxiety. * |
| | **Martinez Monteguado et al. [27]** | Cyberbully victims were more likely to have experienced anxiety. * |
| | **Musharraf et al. [33]** | Weakly significant positive correlation between cybervictimization and anxiety. * |
| | **Na H et al. [17]** | Weakly significant positive correlation between cybervictimization and anxiety. * |
| | **Schenk et al. [16]** | Cyberbully victims experienced significantly greater anxiety. * |
| | **Sam et al. [31]** | Insignificant difference between cyberbullying victims and non-victims. |
| **Suicidal Behaviors** | **Cenat et al. [36]** | Insignificant correlation between cybervictimization and suicidality. |
| | **Chu et al. [34]** | Weakly significant positive correlation between cybervictimization and suicidal ideation. * |
| | **Khine et al. [63]** | Victims were not more likely to have experienced suicidal ideation. |
| | **Martinez Monteguado et al. [27]** | Cyberbully victims were more likely to have experienced suicidal thinking. * |
| | **Medrano et al. [73]** | Weakly significant positive correlation between cybervictimization and suicidal ideation. * |
| | **Schenk et al. [16]** | Cyberbully victims experienced significantly greater suicidal behaviors. * |
| **Psychological Symptoms** | **Cenat et al. [36]** | Weakly significant positive correlation between cybervictimization and psychological distress. * |
| | **Felipe-Castaño, E. et al. [26]** | Weakly significant positive correlation between cybervictimization and psychoticism. * |
| | **Sam et al. [31]** | Cyberbully victims experienced significantly greater psychological symptoms. * |
| | **Schenk et al. [16]** | Insignificant findings between cybervictimization and psychotic subscales. |

(*Continued*)

**Table 2.** (Continued)

| Key Factors | Reference | Results |
|---|---|---|
| **Self-esteem** | **Albikawi et al. [37]** | Cyberbully victims were less likely to have higher self-esteem. * |
| | **Huang et al. [60]** | Insignificant difference between cyberbullying victims and non-victims. |
| | **Na et al. [17]** | Weakly significant negative correlation between cybervictimization and self-esteem. * |
| | **Sam et al. [31]** | Insignificant difference between cyberbullying victims and non-victims. |
| | **Snaychuk, L.A. et al. [12]** | Victims of TFSV had significantly lower self-esteem. * |
| | **Yubero et al. [28]** | Cyberbully victims had significantly lower self-esteem. * |
| | | Very weakly significant negative correlation between cybervictimization and self-esteem. * |
| **Well-being** | **Musharraf et al. [33]** | Very weakly significant negative correlation between cybervictimization and well-being. * |
| | **Anis-ul-Haque et al. [32]** | Individuals with cyberbullying experiences had significantly poorer well-being versus those without. * |
| **Stress** | **Kaur et al. [62]** | Cyberbully victims were more likely to have experienced stress. * |
| | **Martinez Monteguado et al. [27]** | Cyberbully victims were more likely to have experienced stress. * |
| | **Musharraf et al. [33]** | Weakly significant positive correlation between cybervictimization and stress. * |
| **Internet/Social Media Addiction** | **Albikawi et al. [37]** | Cyberbully victims were more likely to have experienced internet addiction. * |
| | **Huang et al. [60]** | Cyberbully victims in social media and online gaming were more likely to have experienced internet addiction. * |
| **Alcohol Use** | **Giumetti et al. [23]** | Insignificant association between cybervictimization and alcohol use. |
| | **Selkie et al. [15]** | Cyberbully victims were not more likely to have experienced alcohol abuse. |

Key

* denotes a significant association.

when evaluating anxiety (p = 0.0415). The remainder of outcomes demonstrated no study publication bias (**S1**–**S7 Figs**).

## Discussion

The current digital landscape has sparked an interest in the widespread mental health impact of cyberbullying. Research, however, has largely focused on adolescents and high school students, resulting in a scarcity of literature regarding university students. Even the handful of studies conducted previously on the topic are by and large unstandardized, with articles varying considerably in terms of participants' sociodemographic characteristics. To our knowledge, ours is the first-ever systematic review on adverse mental health outcomes associated with cyberbullying among university students.

The prevalence of cybervictimization was found to vary across studies from 6.9% to 84.3% [3,24]. This wide variation may be attributed to the lack of a uniform definition of cyberbullying. While the term may seem self-explanatory, cyberbullying encompasses a range of online behaviors, making it difficult to define the term operationally. In addition, a variety of different survey instruments were used to measure cyberbullying across studies, with twelve of the articles employing a self-designed questionnaire [13,14,20,21,29,30,36,38–42]. Moreover, there were differences seen in the amount and frequency of negative exposure that would be required to qualify as cyberbullying. For instance, some articles considered exposures within the last 12 months to meet the criteria of cyberbullying [17], some within the last 6 months [30], and some only in the last 2 months [25]. These discrepancies compound the ambiguity of the term and allow perpetrators to continue their harmful actions without being held accountable.

Students who are targeted may be unsure if their experience qualifies as cyberbullying and thus may not report it. When victims are unable to recognize and report their experience as abuse, they are susceptible to feelings of shame and guilt, which not only takes a toll on their mental health but also often allows victimization to continue [92]. Standardization of the definition of cyberbullying would allow for proper identification of students at risk, as well as more consistent reporting.

Furthermore, literature has demonstrated the impact of cultural contexts on the perception and response to bullying, where western cultures were found to be more forthcoming and receptive towards the entity [93]. These results are not dissimilar to our findings, where certain outcomes such as anger and depressive symptoms were seen to be higher amongst eastern regions [30,32]. Thus, it may be possible that increased awareness and openness to cybervictimization may enable improved coping and reduced development of outcomes in certain geographic regions, however, further research is necessitated to warrant this assumption. To counter the alarming rates of cyberbullying among students, it is also essential for university administrations to design efficient and confidential reporting systems and equip their students with the insight to recognize cyberbullying. This will enable students to report incidents without guilt or fear of criticism.

Our results show that depression and anxiety are the most frequent adverse mental health outcomes reported among cybervictims. This finding is consistent with similar studies conducted amongst adolescents and teenagers [94], which further reaffirms the need for implementing interventions across all age groups. Though interventions have been studied extensively at the adolescent level; there is a significant lack of literature in the setting of higher education [95]. Literature has demonstrated that interventions in the university setting have included interventional videos, zero-tolerance policies, and cyberbullying reporting systems however limited research has been conducted on the effectiveness of such interventions [46,47]. Furthermore, barriers towards intervention access are difficult to address, with studies finding university students to be apprehensive when seeking help due to perceptions of cyberbullying as a "juvenile" issue [96]. As such, it is imperative that further research is conducted to determine systemic barriers towards intervention access, and the impact of a wider array of interventions in the higher education setting. In addition to primary interventions to reduce the prevalence of cyberbullying across universities, targeted psychological interventions can be implemented for individuals who are already experiencing these adverse effects to also allow for tertiary prevention of outcomes. Counselling services have been prevalent amongst institutes of tertiary education, with research demonstrating positive impact towards mental and academic outcomes amongst university students [97,98]. Thus, targeted counselling amongst victims of cyberbullying may enable the development of effective coping strategies, although further research is warranted to substantiate this assumption. This would help alleviate the burden of mental health issues as well as help improve the overall wellbeing among university students.

Our review has several limitations. Firstly, the tools and definitions used to define and measure cyberbullying and mental health outcomes differed across studies and precluded statistical pooling of data. As such, there is considerable heterogeneity across the dataset thus limiting the generalizability of the data and potentially underestimating or overestimating of the prevalences of mental health outcomes. This heterogeneity further signifies the need to establish a uniform framework to identify and counteract cybervictimization. Despite this limitation, however, our study is the first systematic review to evaluate the impact of cyberbullying in developing mental health outcomes, thus highlighting a critical avenue for further research. Secondly, only 22 out of the 32 included articles 11,14–17,20,22,23,25–28,31,33–37,60,62,63,73] were deemed to be of good quality, with the rest being fair or poor, with such

articles failing to validate their study questionnaires. Such lack of validation may result in a lower quality of evidence, due to reduced reliability of results and a potential increase in study bias preventing inference of study outcomes. These findings were manifested in our review, where ten articles were found to be fair or poor, thus increasing the potential for heterogeneity and reduced validity of the conclusions drawn. Similarly, the majority of articles included in this study were of a cross-sectional study design, further preventing accurate inference of causality in the relationship between cybervictimization and mental health outcomes. Due to these limitations, it is possible that the conclusions drawn within this review may be over or under-estimated. As such, it is imperative that further research is conducted emphasizing a longitudinal and robust approach, thus allowing for a more accurate assessment of mental health outcomes. Lastly, elements of recall, social desirability, and publication bias may have been present owing to the cross-sectional design and sensitive nature of the studies. Our results demonstrated that publication bias was only found in one outcome (anxiety), however, due to the limited number of articles evaluating outcomes, it is possible that further publication is present. Thus, it is imperative that great nuance is employed when interpreting our results, however, the studies included in this review were conducted across a wide range of settings, which increases the generalizability of our conclusions.

## Conclusions

In this review, we demonstrate that cyberbullying has an immense psychological burden on university students, increasing the risk of depression, anxiety, stress, and even suicidal behaviour. We recommend that institutions of higher education across the globe enact zero-tolerance policies regarding cyberbullying, implement accessible reporting systems among student bodies, develop anonymous mental health screening programs for students, and provide appropriate psychological care to those who experience cyberbullying.

## Supporting information

**S1 Checklist. This file includes the PRISMA checklist used in the synthesis of the systematic review.**
(DOCX)

**S1 Search strings. This file includes the search strings used in the review of literature.**
(DOCX)

**S1 Table. This file includes the supplementary tables that are relevant to the body of the manuscript.**
(DOCX)

**S1 Data. This file includes the data availability tables that are relevant to data curation in the systematic review.**
(XLSX)

**S1 Fig. This file includes a supplementary figure relevant to the main body of the manuscript.**
(TIF)

**S2 Fig. This file includes a supplementary figure relevant to the main body of the manuscript.**
(TIF)

**S3 Fig. This file includes a supplementary figure relevant to the main body of the manuscript.**
(TIF)

**S4 Fig. This file includes a supplementary figure relevant to the main body of the manuscript.**
(TIF)

**S5 Fig. This file includes a supplementary figure relevant to the main body of the manuscript.**
(TIF)

**S6 Fig. This file includes a supplementary figure relevant to the main body of the manuscript.**
(TIF)

**S7 Fig. This file includes a supplementary figure relevant to the main body of the manuscript.**
(TIF)

## Author Contributions

**Conceptualization:** Aahan Arif, Muskaan Abdul Qadir, Russell Seth Martins, Hussain Maqbool Ahmed Khuwaja.

**Data curation:** Aahan Arif, Muskaan Abdul Qadir.

**Methodology:** Aahan Arif, Russell Seth Martins, Hussain Maqbool Ahmed Khuwaja.

**Supervision:** Hussain Maqbool Ahmed Khuwaja.

**Writing – original draft:** Aahan Arif, Muskaan Abdul Qadir.

**Writing – review & editing:** Russell Seth Martins, Hussain Maqbool Ahmed Khuwaja.

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
