## [Decision Letter · Decision Letter 0]

12 Jun 2024

PMEN-D-24-00118

The Impact of Cyberbullying on Mental Health Outcomes amongst University Students: A Systematic Review

PLOS Mental Health

Dear Dr. Khuwaja,

Thank you for submitting your manuscript to PLOS Mental Health. After careful consideration, we feel that it has merit but does not fully meet PLOS Mental Health’s publication criteria as it currently stands. Therefore, we invite you to submit a revised version of the manuscript that addresses the points raised during the review process.

I have read your submission by myself and agree with the comments. Therefore, I hope you can revise it accordingly.

Please submit your revised manuscript by . If you will need more time than this to complete your revisions, please reply to this message or contact the journal office at mentalhealth@plos.org. Please include the following items when submitting your revised manuscript:

We look forward to receiving your revised manuscript.

Kind regards,

Wenjie Duan, Ph.D.

Academic Editor

PLOS Mental Health

Journal Requirements:

Additional Editor Comments (if provided):

Reviewers' comments:

Reviewer's Responses to Questions

**Comments to the Author**

1. Does this manuscript meet PLOS Mental Health’s publication criteria? Is the manuscript technically sound, and do the data support the conclusions? The manuscript must describe methodologically and ethically rigorous research with conclusions that are appropriately drawn based on the data presented.

Reviewer #1: Yes

2. Has the statistical analysis been performed appropriately and rigorously?

Reviewer #1: Yes

3. Have the authors made all data underlying the findings in their manuscript fully available (please refer to the Data Availability Statement at the start of the manuscript PDF file)?

Reviewer #1: Yes

4. Is the manuscript presented in an intelligible fashion and written in standard English?

Reviewer #1: Yes

5. Review Comments to the Author

Reviewer #1: The manuscript titled is a comprehensive effort to synthesize the existing research on the psychological effects of cyberbullying among university students. However, there are several areas where the study could be improved or where limitations should be acknowledged more explicitly:

The manuscript highlights the lack of a uniform definition of cyberbullying across studies, which may have led to variability in findings. This is a critical limitation as it affects the comparability of study results and may inflate or deflate the prevalence rates reported.

Many of the studies included in the review are cross-sectional, which limits the ability to infer causality between cyberbullying and mental health outcomes. The review could benefit from a discussion on the need for longitudinal studies to better understand the causal pathways.

While the manuscript notes that the majority of articles were of good quality, there are still several that were of fair or poor quality. This could potentially bias the overall findings of the review. A more thorough discussion on how these quality issues might impact the review's conclusions would be beneficial.

The studies included span a range of geographic and cultural contexts, which can influence both the experience of cyberbullying and its impact on mental health. The review could strengthen its analysis by more deeply exploring how cultural differences affect the generalizability of the findings.

The review does not discuss the possibility of publication bias, where studies with significant findings are more likely to be published than those with null results. Addressing this could involve a statistical assessment of publication bias to ensure the robustness of the review's conclusions.

While the review recommends interventions, there is limited discussion on the effectiveness of existing cyberbullying interventions, particularly those targeted at university students. An analysis of intervention studies could provide valuable insights into what works and what doesn’t in preventing or mitigating the effects of cyberbullying.

The review mentions that all data are included in the published article, but it does not discuss the availability of raw data for reanalysis or the reproducibility of the results, which are important for verifying and building on research findings.

6. PLOS authors have the option to publish the peer review history of their article (what does this mean?). If published, this will include your full peer review and any attached files.

**Do you want your identity to be public for this peer review?** For information about this choice, including consent withdrawal, please see our Privacy Policy.

Reviewer #1: No

---

## [Decision Letter · Decision Letter 1]

2 Oct 2024

The Impact of Cyberbullying on Mental Health Outcomes amongst University Students: A Systematic Review

PMEN-D-24-00118R1

Dear Mr Khuwaja,

We are pleased to inform you that your manuscript 'The Impact of Cyberbullying on Mental Health Outcomes amongst University Students: A Systematic Review' has been provisionally accepted for publication in PLOS Mental Health.

Best regards,

Wenjie Duan, Ph.D.

Academic Editor

PLOS Mental Health

Reviewer Comments (if any, and for reference):

Reviewer's Responses to Questions

**Comments to the Author**

1. If the authors have adequately addressed your comments raised in a previous round of review and you feel that this manuscript is now acceptable for publication, you may indicate that here to bypass the “Comments to the Author” section, enter your conflict of interest statement in the “Confidential to Editor” section, and submit your "Accept" recommendation.

Reviewer #1: All comments have been addressed

2. Does this manuscript meet PLOS Mental Health’s publication criteria? Is the manuscript technically sound, and do the data support the conclusions? The manuscript must describe methodologically and ethically rigorous research with conclusions that are appropriately drawn based on the data presented.

Reviewer #1: Yes

3. Has the statistical analysis been performed appropriately and rigorously?

Reviewer #1: N/A

4. Have the authors made all data underlying the findings in their manuscript fully available (please refer to the Data Availability Statement at the start of the manuscript PDF file)?

Reviewer #1: Yes

5. Is the manuscript presented in an intelligible fashion and written in standard English?

Reviewer #1: Yes

6. Review Comments to the Author

Reviewer #1: The revisions made in response to the previous comments have been addressed adequately. The methodology, results, and limitations are now clearly presented, and the manuscript is significantly improved. No further changes are needed.

7. PLOS authors have the option to publish the peer review history of their article (what does this mean?). If published, this will include your full peer review and any attached files.

**Do you want your identity to be public for this peer review?** For information about this choice, including consent withdrawal, please see our Privacy Policy.

Reviewer #1: No
